# Influences of CO₂ Absorption under Ultrasonic Vibration and Water-Reducer Addition on the Rheological Properties of Cement Paste

**Lili Liu [1], Yongsheng Ji [2],\*, Longhai Li [1] and Jie Zhang [2]**

[1] School of Mechanical and Electrical Engineering, Xuzhou University of Technology, Xuzhou 221000, China; dxwlll@sina.com (L.L.); longhaicumt@163.com (L.L.)

[2] Jiangsu Key Laboratory Environmental Impact and Structural Safety in Engineering, China University of Mining and Technology, Xuzhou 221116, China; jys15152110206@sina.com

\* Correspondence: jiyongsheng@cumt.edu.cn; Tel.: +86-516-8399-5295

**Abstract:** Research shows that ultrasonic vibratory stirring can effectively increase $CO_2$ absorption by cement slurry. However, with the increase in $CO_2$ absorption, the fluidity of slurry begins to decrease. Adding water reducer to fresh cement paste can improve its fluidity. In order to reveal the influences of ultrasonic vibration and water-reducer addition on the rheological properties of cement pastes after absorbing various amounts of $CO_2$, changes in the rheological properties of yield stress and plastic viscosity (PV) were analysed. The results show that ultrasonic vibration can effectively increase the shear stress and PV of cement paste. Moreover, shear stress and PV are positively related to the $CO_2$ absorption amount. Meanwhile, a new rheological model of cement paste carbonated under ultrasonic vibration was established based on the basic principles of rheology. Microstructural changes in cement paste before and after water-reducer addition were observed by scanning electron microscopy (SEM). A microrheological model of cement paste carbonated under ultrasonic vibration and with water reducer added was constructed. It describes the influencing mechanisms of ultrasonic vibration and water-reducer addition on the rheological properties of carbonated cement paste. Next, a molecular model was constructed in which $CO_2$ was added into a C-S-H gel. Changes in intermolecular repulsion in the $CO^{2+}$ C-S-H gel structure and in the $CO^{2-}$ water-reducer molecular structure were analysed. Finally, the rheological mechanism was further analysed in terms of the dispersion effect of the C-S-H gel. The results will play a major role in improving the fluidity of cement paste.

**Keywords:** fresh cement paste; ultrasonic agitation; absorb $CO_2$; polycarboxylate superplasticiser; rheological property

## 1. Introduction

The greenhouse effect is caused by excessive $CO_2$ emissions and poses many environmental hazards. Global warming has become one of the world's top 10 global environmental problems [1,2]. Nowadays, many methods exist to decrease global $CO_2$ emissions, such as changing energy structures, chemical absorption and blocked curing [3]. However, all of these methods have certain technological defects. They all incur relatively high costs and cannot solve carbon-emission problems within a short period of time [4,5]. Some studies have pointed out that the cement in concrete can produce a significant amount of $Ca(OH)_2$ during the hydration process (20–30% of the total hardened cement paste), which largely exists in the pores of hardened cement pastes in crystal form [6–8]. In solution, $Ca(OH)_2$ can react with $CO_2$ extremely easily to produce $CaCO_3$. As a result, concrete has considerable potential for use as a $CO_2$ absorbent [9–11].

At present, research on the absorption of $CO_2$ by freshly mixed slurry has achieved some progress. Our research group has developed mechanical and ultrasonic vibratory-stirring devices to increase $CO_2$ absorption by cement slurry [12–14]. The research shows

that ultrasonic vibratory stirring can effectively increase $CO_2$ absorption by cement slurry under the condition of mechanical stirring (at a certain stirring rate and water–cement ratio) [12,15]. Furthermore, the mechanical properties of the cement slurry are improved after absorbing $CO_2$. However, the fluidity of the fresh cement paste is negatively related to the amount of $CO_2$ absorption ($CO_2$ AA). Although ultrasonic vibration can improve the fluidity of fresh cement paste, the paste solidifies as the $CO_2$ AA increases and the paste eventually loses its working performance [16–18]. Adding a water reducer to fresh cement paste can improve its fluidity, thus regaining its working performance [19–21]. However, the nature of the fluidity change after adding water reducer under ultrasonic vibration remains unclear.

Therefore, this paper analyses the rheological principles and internal microstructural changes of fresh slurry with an added water-reducing agent. Further, it establishes molecular models and conducts dynamic analysis to deeply explore the mechanisms and effects of ultrasonic vibration.

## 2. Basic Theory of Rheology

The rheological models most commonly used to describe the fluidity of fresh cement paste are the Bingham and Herschel–Bulkley (H-B) models. In this investigation, the steady-state rheological parameters of fresh cement paste were obtained according to an RS-SST rheometer (Brookfield Company), then the values of these parameters were fitted based on Bingham, modified Bingham (M-B) and H-B models to analyse the rheological properties of fresh cement paste after $CO_2$ absorption. The corresponding basic rheological models are briefly introduced as follows.

(1) Bingham model:

The Bingham model (Expression (1)) is a relatively common rheological model used to describe the fluidity of fresh cement-based materials. If the external forces that the fluid bears are less than the yield stress, the fluid is characterised by plastic flow; otherwise, the fluid is characterised by viscous flow.

$$\tau = \tau_0 + \eta\gamma \tag{1}$$

(2) Modified Bingham (MB) model:

The Bingham model is an ideal model, while the MB model can more accurately characterise the rheology of cement-based materials. Its expression is

$$\tau = \tau_0 + \eta\gamma + c\gamma^2 \tag{2}$$

(3) Herschel–Bulkley (H-B) model

The H-B model is often used to describe the fluidity of fresh concrete, cement pastes and particle-containing suspensions. Compared with the Bingham model, the H-B model is more accurate. According to the H-B model, the plastic viscosity (PV) of fluid is mainly influenced by shear stress, which is mainly modelled by variations in a power exponent, usually denoted as $n$. When $n < 1$, the shear stress decreases with increases in PV, thus leading to shear thinning. When $n = 1$, the fluid is characterised as a Bingham fluid. When $n > 1$, lower viscosity is obtained with higher shear stress at a lower shear rate, which results in shear thinning; and higher viscosity is obtained with higher shear stress at a higher shear rate, which results in shear thickening. The H-B model is expressed as follows:

$$\tau = \tau_0 + K\gamma^n \tag{3}$$

Rheological curves of the three rheological models are shown in Figure 1.

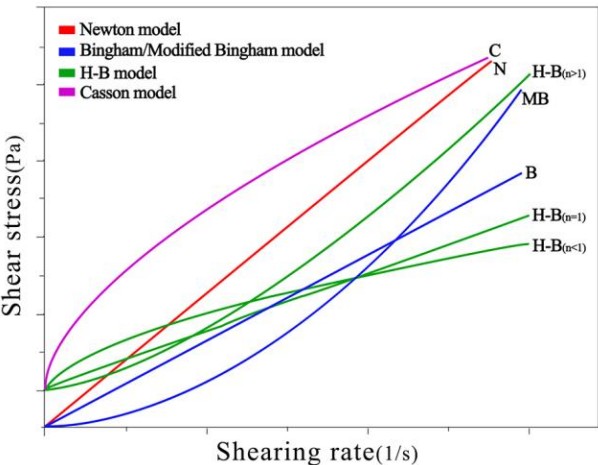

**Figure 1.** Existing rheological models of fluids. Three H-B models are shown with different values of *n*.

## 3. Test Materials and Methods

### 3.1. Raw Materials

The cement used in this study was P·O 42.5 cement produced by the Xuzhou Zhonglian Cement Group. The mean grain size, density, standard-consistency water requirement, fineness (0.08 mm square-hole sieve) and specific surface area were 14.813% Xav (μm), 3.14 g/cm$^3$, 28.1%, 1.02% and 3300 cm$^2$/g, respectively. The specific chemical composition and mineral composition are listed in Tables 1 and 2. The admixture is polycarboxylate superplasticiser (PS) in Table 3. The $CO_2$ was high-purity (≥99.5%) $CO_2$ produced by a special gas plant in Xuzhou. Tap water was also used.

**Table 1.** Chemical composition of the P·O 42.5 cement.

| Chemical Composition | $SiO_2$ | $Al_2O_3$ | $Fe_2O_3$ | CaO | MgO | f-CaO | Loss |
|---|---|---|---|---|---|---|---|
| Content (%) | 22.1 | 5.34 | 3.44 | 65.33 | 2.11 | 0.39 | 0.13 |

**Table 2.** Mineral composition of the P·O 42.5 cement.

| Composition | $C_3S$ | $C_2S$ | $C_3A$ | $C_4AF$ |
|---|---|---|---|---|
| Content (%) | 54.04 | 22.84 | 8.39 | 10.42 |

**Table 3.** Properties of PS.

| Appearance | Solid Content (%) | Density (g/mL) | pH | Chloride Ion Content (%) | Alkali Content (%) | Water Reduction Rate (%) |
|---|---|---|---|---|---|---|
| Light brown liquid | 25 ± 2 | 1.07 ± 0.02 | 6~8 | 8.39 ≤ 0.02 | ≤0.2 | 25~45 |

### 3.2. Test Equipment

Two $CO_2$-absorption devices were manufactured to investigate the influences of ultrasonic-vibration agitation and PS addition on the rheological properties of $CO_2$ AA of fresh cement paste. One was a mechanical agitation device (Figure 2a) and the other was an ultrasonic-vibration agitation tank. The manufacturing process is introduced as follows. Firstly, the ultrasonic-vibration agitation tester with variable frequencies was connected to the original $CO_2$-absorption device. Next, a transducer, ultrasonic power supply and qualified amplitude transformer were selected according to vibration amplitude and frequency requirements of the vibration system. Finally, the ultrasonic power supply, transducer, amplitude transformer and agitation tank were connected (Figure 2b).

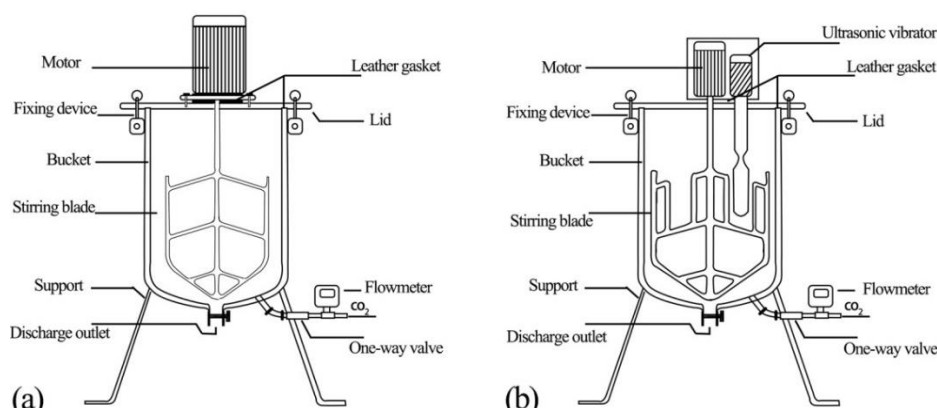

**Figure 2.** Schematic diagrams of agitation devices (Reprinted with permission from Ref. [21]. 2021, Constr. Build. Mater, Elsevier.): (**a**) Mechanical agitation device; (**b**) ultrasonic agitation devices.

A rotational rheometer (RS-SST, Brookfield) was used to test the rheological properties of the cement pastes. The propeller rotor was a model VT-60-30. We used an ISO-standard consistency and setting-time tester for cement paste produced by Cangzhou Kexing Instrument Co. Ltd. The maximum sliding stroke of the testing cone was 70 mm. The diameter of the dispatching rod was $10 \pm 0.05$ mm and the diameter of the needle was $1.13 \pm 0.05$ mm.

### 3.3. Preparation of Net Cement-Paste Samples

The water–cement ratio of the net cement-paste samples was 0.6. The cement content was 800 g and the water content was 400 g. The cement and water mixture was poured into the agitation vessel and the valve of the $CO_2$ tank was opened to inject the corresponding mass of $CO_2$. After the $CO_2$ was completely absorbed by the cement paste, it was stirred slowly for 2 min. The stirring was stopped for 15 s and then the mixture was quickly stirred again for 1 min 45 s.

### 3.4. Rheological Property Test of the Net Cement Paste

The shear stress and apparent viscosity of cement pastes with different $CO_2$ AA were tested by changing the shearing rate of the rheometer. The detailed test process is shown in Figure 3. In the pre-shear stage (0–30 s), the shear rate increased linearly from 0 to 50 s$^{-1}$. Later, the shear rate was constant (30–90 s). At 90–120 s, the shear rate decreased linearly to 0 and the pre-shear stage finished. Subsequently, the cement paste was left to stand for 60 s (120–180 s). In the first half of the test stage (180–360 s), the shear rate increased linearly from 0 to a maximum of 200 s$^{-1}$. In the second half of the test stage, the shear rate decreased sharply from 200 s$^{-1}$ to 0. Data were recorded every 1 s throughout the test process.

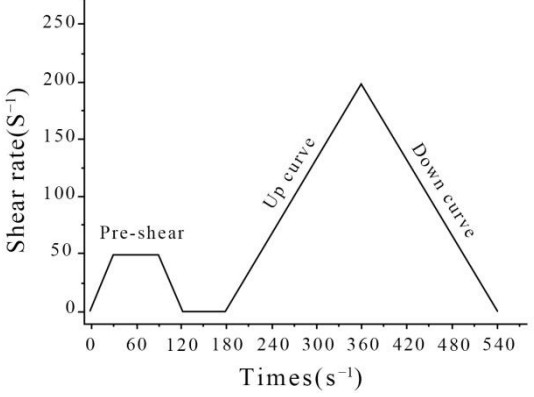

**Figure 3.** Rheological test data.

Since the steady state of non-Newtonian fluid stress is relatively short when the shear rate decreases, the tested shear-stress curve is more stable. This also implies that the test results can reflect the rheological properties of cement paste better. On this basis, variations in the descending parts of the shear-rate curves were chosen for analysis. In this study, data were analysed using Origin software, which also provided the rheological equation and corresponding rheological parameters.

## 4. Test Contents and Methods

### 4.1. Influences of Ultrasonic Vibration on the Rheological Properties of Cement Paste after $CO_2$ Absorption

After $CO_2$ absorption, the cement pastes were tested under mechanical agitation (Group A) or ultrasonic agitation (Group B). The stirring rate was set to $210 \pm 5$ r/min. The ultrasonic frequency was 20 kHz and the corresponding $CO_2$ AAs were 0%, 0.44%, 0.88%, 1.32%, 1.76% and 2.20% of the cement mass. Cement-paste samples with a water–cement ratio (W:C) of 0.5 were used in the tests. The volumes of cement and water consumed were 800 g and 400 g, respectively. Weighed amounts of cement and water were added to an agitator and stirred. Meanwhile, a corresponding mass of $CO_2$ was supplied to the agitator. The agitator was turned off after the $CO_2$ was completely absorbed by the cement paste. The agitation program comprised 4 min agitation, slow stirring for 2 min, a 15 s pause, then fast stirring for 105 s. In this process, the $CO_2$ AA was increased at a rate of 0.44% and pastes were collected in a 500 mL beaker before each increase. All beakers were installed with appropriate rotors to test the rheological properties of the fresh cement pastes. Based on these tests, the influences of ultrasonic vibration on the rheological properties of cement pastes after $CO_2$ absorption were analysed.

### 4.2. Influences of Water Reducer on the Rheological Properties of Cement Paste after $CO_2$ Absorption

The test was divided into two groups (C and D). Group C used cement paste without PS under ultrasonic agitation, while Group D used cement paste with 0.25% PS under ultrasonic agitation. The stirring rate was set to $210 \pm 5$ r/min and the ultrasonication frequency was 20 kHz. In the tests, the influences of PS addition on the rheological properties of cement paste under ultrasonic agitation were investigated [22].

### 4.3. Morphological Features of Products Precipitated from Cement Paste with Water Reducer Added after $CO_2$ Absorption

The test was divided into two groups (E and F). Group E used fresh cement paste with a 2.20% $CO_2$ AA (of the cement mass) under ultrasonic agitation. The W:C was 0.5 and the ultrasonic frequency was 20 kHz. Group F used fresh cement paste with 0.25% PS (of the cement mass) under ultrasonic agitation. These two fresh cement pastes were poured into 40 mm cubic moulds and compacted by vibration; then, the samples were cured under standard conditions for 12 h to prepare cement-paste samples. At this time, the strength of the cement-paste samples was initially established, which was conducive to sample preparation. At this time, the hydration reaction in the cement had only just begun and its internal structure was relatively loose. This made it a good time to observe the products of the reaction of cement-hydration products with $CO_2$.

The cement-paste samples were made into 1 mm-thick pieces, which were then immersed in absolute ethanol for 48 h to stop the hydration of the cement. The pieces were dried in a constant-temperature blast oven at 65 °C for 24 h, and then placed in an ion-sputtering apparatus for surface gold spraying. SEM and energy spectrum analysis were then performed.

## 5. Test Materials and Equipment

*5.1. Effects of CO$_2$ Absorption on the Rheological Properties of Cement Paste*

(1) Rheological curves of cement paste after CO$_2$ absorption

(a) Mechanical vibration

The rheological curves of the cement pastes after absorbing CO$_2$ under mechanical vibration are shown in Figure 4a. The shear stresses of the net cement pastes in different groups are summarised in Table 4. It can be seen from Figure 4a and Table 4 that the initial shear stress of the net cement paste under mechanical vibration is 7.56 Pa. At shear rates <120 L/s, the shear stress increases slowly with the shear rate. At shear rates >120 L/s, the shear thickening increases quickly and the shear stress increases in a straight line. At CO$_2$ AA = 0.44%, the paste-thickening phenomenon occurs and the initial shear stress increases to 17.5 Pa. With increases in shear rate, the shear stress increases slowly and the rate of increase declines to some extent. At CO$_2$ AA = 0.88%, the initial shear stress increases further to 25.1 Pa and the trend is very similar to that at 0.44%. At CO$_2$ AA = 1.32%, the shearing resistance continues to increase, while the initial shear stress increases to 29.6 Pa. Under this circumstance, the rheological curve rises at a slow rate. At CO$_2$ AA = 1.76%, the thickening phenomenon is more obvious. The trend when the initial shear stress is increased to 46.5 Pa is similar to that at 1.32%. Shear thickening becomes more obvious at CO$_2$ AA = 2.20%. The initial shear stress reaches 77.6 Pa and the rheological curve has a parabolic shape.

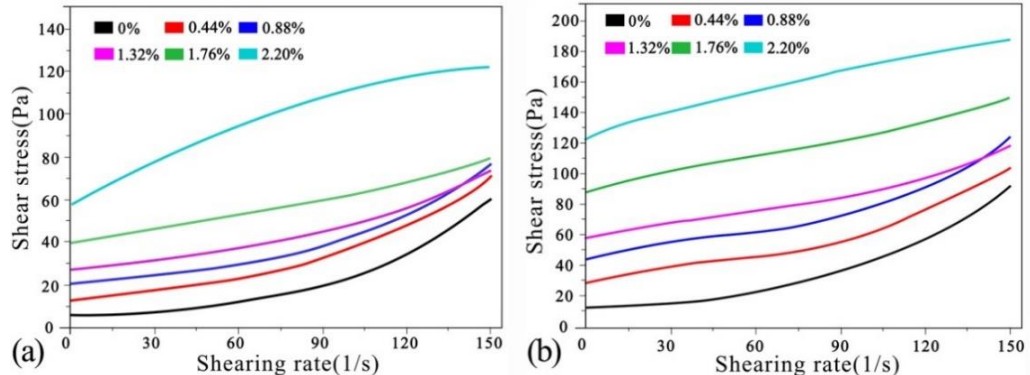

**Figure 4.** Rheological curves of cement pastes after CO$_2$ absorption under (**a**) mechanical and (**b**) ultrasonic agitation.

**Table 4.** Shear stresses of cement paste after CO$_2$ absorption under mechanical agitation (Pa).

| CO$_2$ AA/% | 30 L/s | 60 L/s | 90 L/s | 120 L/s | 150 L/s |
|---|---|---|---|---|---|
| 0 | 7.56 ± 1.35 | 12.3 ± 3.45 | 20.1 ± 5.96 | 34.5 ± 4.53 | 60.1 ± 6.75 |
| 0.44 | 17.5 ± 3.54 | 22.5 ± 4.89 | 32.5 ± 7.32 | 48.5 ± 4.14 | 74 ± 7.03 |
| 0.88 | 25 ± 4.79 | 30 ± 6.24 | 38 ± 6.46 | 55 ± 6.29 | 77.5 ± 8.21 |
| 1.32 | 29.6 ± 5.21 | 35 ± 6.35 | 45 ± 8.23 | 57 ± 6.31 | 74.2 ± 8.13 |
| 1.76 | 46.5 ± 8.18 | 53.6 ± 8.2 | 60 ± 8.86 | 68.2 ± 8.92 | 80 ± 8.88 |
| 2.20 | 77.6 ± 8.06 | 94.1 ± 8.12 | 107.5 ± 10.11 | 117.5 ± 10.36 | 112.3 ± 10.23 |

(b) Ultrasonic vibration stirring

The rheological curves of cement pastes after CO$_2$ absorption under ultrasonic vibration are shown in Figure 4b. The shear stresses of the net cement pastes of all groups are summarised in Table 5. It can be seen from Figure 4b and Table 5 that in the initial stage of the shear rate of net cement pastes under ultrasonic vibration, the shear stress is 12.8 Pa. At shear rates of 0–90 L/s, the shear stress increases slowly. At shear rates >90 L/s, shear thickening increases in a straight manner. At CO$_2$ AA = 0.44%, the initial shear stress increases to 40.2 Pa and the paste thickening increases obviously. At shear rates of 0–120 L/s, the shear stress increases slowly. The thickening accelerates at shear rates

>120 L/s, resulting in a linear increase in shear stress. At $CO_2$ AA = 0.88%, the initial shear stress increases to 55.1 Pa and the curve is similar to that at 0.44%. When the $CO_2$ AA is increased to 1.32%, the initial shear stress increases to 69.1 Pa. Under this circumstance, the rheological curves increase at a steady rate without obvious growth. At $CO_2$ AA = 1.76%, the thickening phenomenon is more significant and the initial shear rate increases to 105 Pa. At $CO_2$ AA = 2.20%, shear thickening continues to increase and the rheological curves also increase slowly. (Note: N = 10, mean ± S.D. N presents number of samples).

**Table 5.** Shear stresses of cement pastes after $CO_2$ absorption under ultrasonic agitation (Pa).

| $CO_2$ AA/% | 30 L/s | 60 L/s | 90 L/s | 120 L/s | 150 L/s |
|---|---|---|---|---|---|
| 0 | 12.8 ± 3.4 | 22.5 ± 3.1 | 37.5 ± 6.4 | 57.2 ± 6.8 | 91.8 ± 7.2 |
| 0.44 | 40 ± 5.7 | 46.2 ± 4.0 | 55 ± 7.2 | 77.2 ± 7.2 | 103.2 ± 8.2 |
| 0.88 | 55 ± 5.2 | 62.5 ± 7.6 | 74 ± 8.2 | 92.5 ± 8.1 | 124.5 ± 8.1 |
| 1.32 | 69.1 ± 6.0 | 77.2 ± 8.3 | 85 ± 8.1 | 97.4 ± 8.3 | 119.6 ± 9.2 |
| 1.76 | 105 ± 9.8 | 111.5 ± 9.2 | 122.1 ± 9.1 | 135.2 ± 10.1 | 150. ± 10.2 |
| 2.20 | 140.4 ± 12.6 | 154.5 ± 11.4 | 167.5 ± 12.9 | 177.5 ± 12.1 | 187.4 ± 13.8 |

(c) Analysis of shear stress in cement pastes after $CO_2$ absorption

Firstly, the shear stresses of cement pastes before and after $CO_2$ absorption under mechanical vibration were compared. At $CO_2$ AA = 0.44%, shear thickening was obvious at shear rates of 30 L/s, 60 L/s, 90 L/s, 120 L/s and 150 L/s. The shear stresses of cement paste were 131.5%, 82.9%, 61.7%, 40.6% and 23.6% higher before $CO_2$ absorption than afterwards. Moreover, the shear thickening of cement paste became more obvious with increases in $CO_2$ AA and was accompanied by continuous increases in shear stress.

Second, compared with mechanical agitation, the shear stresses of the cement slurries without $CO_2$ input under ultrasonic agitation were 69%, 83%, 87%, 66% and 53% higher at shear rates of 30 L/s, 60 L/s, 90 L/s, 120 L/s and 150 L/s, respectively. At $CO_2$ AA = 0.44%, the shear stresses of the cement slurry were 128.6%, 105.3%, 69%, 59.2% and 39.5% higher, respectively. At $CO_2$ AA = 0.88%, the shear stresses of the cement pastes under ultrasonic agitation were 120%, 108.3%, 94.7%, 68.2% and 60.6% higher compared to those under mechanical agitation. At $CO_2$ AA = 1.32%, the shear stresses of cement pastes under ultrasonic agitation were increased by 133.4%, 120.6%, 88.9%, 70.2% and 61.2%, respectively. At $CO_2$ AA = 1.76%, the shear stresses of cement pastes under ultrasonic agitation were increased by 125.8%, 108%, 103.5%, 98.2% and 87.5%, respectively. At $CO_2$ AA = 2.20%, the shear stresses of cement pastes under ultrasonic agitation were increased by 80.9%, 64.2%, 55.8%, 51.1% and 66.9%, respectively.

Based on the above comparison, the shear stress of cement paste was significantly increased by $CO_2$ absorption under mechanical stirring. Meanwhile, the shear stress was higher after ultrasonic vibration than mechanical vibration, indicating that ultrasonic vibration can more effectively increase the shear stress of carbonised cement paste. Comparison of the rheological curves in Figures 4 and 5 shows that after $CO_2$ absorption under both types of vibration, the cement pastes generally showed positive relationships of shear stress with $CO_2$ AA and shear rate. All curves increase at low rates and some present linear trends after reaching a certain shear rate. Although there are similarities between the curves, they change differently.

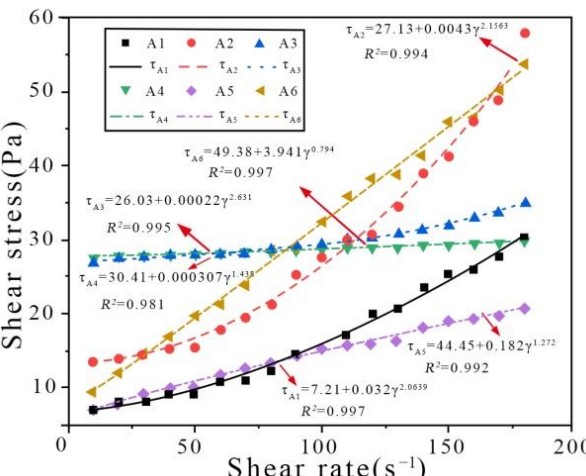

**Figure 5.** Rheological models of cement slurry carbonated under mechanical agitation. Note: A1–A6 represent the models at $CO_2$ AA = 0%, 0.44%, 0.88%, 1.32%, 1.76% and 2.20%, respectively.

### 5.2. Rheological Models of Cement Paste after $CO_2$ Absorption

An accurate rheological model is the premise for evaluating the rheological characteristics of cement paste. A rheological model of cement paste after $CO_2$ absorption was derived from a comparative analysis of the rheological curves (Figure 4) and the rheological models of several fluids (Figure 1) [23–25].

(1) Rheological models of cement paste under mechanical vibration

It can be seen from Figure 4a that under mechanical vibration, shear stress is positively related to shear rate, and the rheological curve shows three stages of increase. The rheological curve of cement paste carbonated under mechanical vibration (Figure 1) has a similar shape as the H-B model; hence, this model was selected for fitting to the observed data.

The rheological parameters of cement pastes carbonated under mechanical vibration were fitted using the H-B rheological model. The fitted-model parameters of cement pastes carbonated under mechanical vibration are shown in Figure 5. It can be clearly seen from the fitting data that the power exponent ($n$) in the parameters in Table 6 meets the H-B rheological model, $n \leq 1$ and $n > 1$. It approaches to the H-B model ($n > 1$) at $CO_2$ AA = 0.44% and 0.88%. It approaches to the H-B model ($n = 1$) at $CO_2$ AA = 1.32% and 1.76%. However, it conforms to the H-B model ($n < 1$) at $CO_2$ AA = 2.20%. Specifically, the R2 coefficient is close to 1 in all cases, indicating accurate modelling. In short, the rheology of cement pastes carbonated under mechanical vibration conforms to the H-B rheological model.

**Table 6.** Rheological curve-fitting parameters of cement slurry carbonated under mechanical vibration.

| No. | $CO_2$ AA | $\tau_0$ (Pa) | K | $n$ | $R^2$ |
|-----|-----------|---------------|---|-----|-------|
| A1 | 0% | 7.21 | 0.0032 | 2.0639 ($n > 1$) | 0.997 |
| A2 | 0.44% | 21.13 | 0.0043 | 2.1563 ($n > 1$) | 0.994 |
| A3 | 0.88% | 26.03 | 0.00022 | 2.631 ($n > 1$) | 0.995 |
| A4 | 1.32% | 30.41 | 0.00307 | 1.438 ($n \approx 1$) | 0.981 |
| A5 | 1.76% | 44.45 | 0.182 | 1.272 ($n \approx 1$) | 0.992 |
| A6 | 2.20% | 49.38 | 3.941 | 0.794 ($n < 1$) | 0.997 |

(2) Rheological models of cement paste carbonated under ultrasonic vibration

In this study, the rheological characteristics of cement paste carbonated under ultrasonic vibration were modelled based on the Bingham, M-B and H-B models (Figure 6). It can be seen from Figure 6 and Table 7 that all three models have low accuracy with $R^2$ coefficients < 0.8. Therefore, none of these three models can accurately characterise the rheological characteristics.

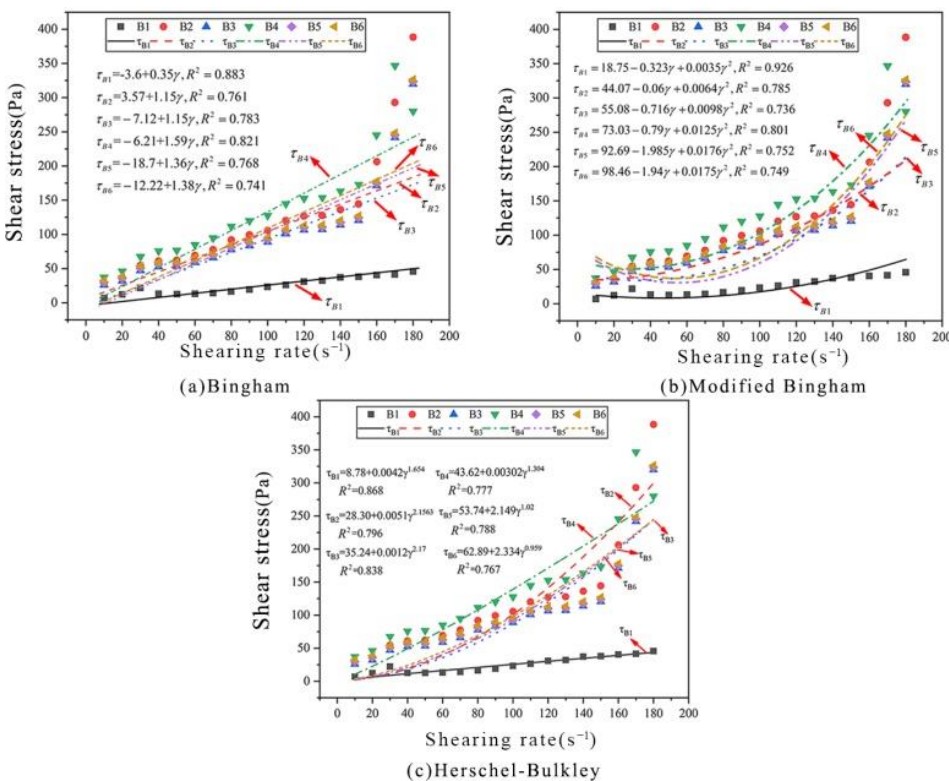

**Figure 6.** Rheological models of cement slurry carbonated by ultrasonic vibration. Note: B1–B6 represents rheology at $CO_2$ AA = 0%, 0.44%, 0.88%, 1.32%, 1.76% and 2.20%, respectively.

**Table 7.** Rheological-model parameters of cement slurry carbonated under ultrasonic vibration.

| Sample | Bingham Model | $R^2$ | MB Model | $R^2$ | H-B Model | $R^2$ |
|---|---|---|---|---|---|---|
| $B_1$ | $\tau_{B1} = 3.6 + 0.35\gamma$ | 0.883 | $\tau_{B1} = 18.75 - 0.323\gamma + 0.0035\gamma^2$ | 0.926 | $\tau_{B1} = 8.78 + 0.0042\gamma^{1.654}$ | 0.868 |
| $B_2$ | $\tau_{B2} = 3.57 + 1.15\gamma$ | 0.761 | $\tau_{B2} = 44.07 - 0.06\gamma + 0.0064\gamma^2$ | 0.785 | $\tau_{B2} = 28.30 + 0.0051\gamma^{2.1563}$ | 0.796 |
| $B_3$ | $\tau_{B3} = 7.12 + 1.15\gamma$ | 0.783 | $\tau_{B3} = 55.08 - 0.716\gamma + 0.0098\gamma^2$ | 0.736 | $\tau_{B3} = 35.24 + 0.0012\gamma^{2.17}$ | 0.838 |
| $B_4$ | $\tau_{B4} = 6.21 + 1.59\gamma$ | 0.812 | $\tau_{B4} = 73.03 - 0.79\gamma + 0.0125\gamma^2$ | 0.801 | $\tau_{B4} = 43.62 + 0.00302\gamma^{1.304}$ | 0.777 |
| $B_5$ | $\tau_{B5} = 18.7 + 1.36\gamma$ | 0.768 | $\tau_{B5} = 92.69 - 1.985\gamma + 0.0176\gamma^2$ | 0.752 | $\tau_{B5} = 53.74 + 2.149\gamma^{1.02}$ | 0.788 |
| $B_6$ | $\tau_{B6} = -12.22 + 1.38\gamma$ | 0.741 | $\tau_{B6} = 98.46 - 1.94\gamma + 0.0175\gamma^2$ | 0.749 | $\tau_{B6} = 62.89 + 2.334\gamma^{0.959}$ | 0.767 |

*5.3. Construction of Rheological Models of Cement Paste Carbonated under Ultrasonication*

(1) Rheological models of cement paste carbonated under ultrasonication

Based on the above analysis, the rheological curves of cement pastes carbonated under ultrasonication do not all conform to the traditional Bingham, M-B and H-B models as well as those carbonated under mechanical vibration. However, it can be concluded from the $R^2$ values although the fitted curves have low accuracy, they do have some of the characteristics of the rheological models [26–28]. This is because the above models are ideal and cannot characterise the actual ultrasonication characteristics. On this basis, a practical rheological model that is applicable to cement-based materials carbonated under ultrasonic vibration is proposed by combining characteristics and advantages of the above three models:

$$\tau = \tau_0 + \varepsilon * \gamma + \eta * \gamma^2 + \xi * \gamma^3 + \varsigma * \gamma^4 \tag{4}$$

where $\tau$ is the shear stress; $\tau_0$ is the yield stress; $\gamma$ is the shear rate; and $\varepsilon$, $\eta$, $\xi$ and $\varsigma$ are constants.

(2) Applications of rheological models of cement paste carbonated by ultrasonication

To judge whether the rheological model of carbonated cement paste is applicable to the actual rheological model of cement-based materials carbonated under ultrasonic vibration,

rheological data under different $CO_2$ AA values were modelled again. The results are shown in Figure 7 and Table 8. Clearly, the $R^2$-values are >0.9 for all rheological curves of cement pastes with different $CO_2$ AA values, indicating that these curves meet the fitting requirements well. This proves that the rheological model of carbonated cement paste is appropriate for the actual rheological characteristics of cement-based materials carbonated under ultrasonic vibration.

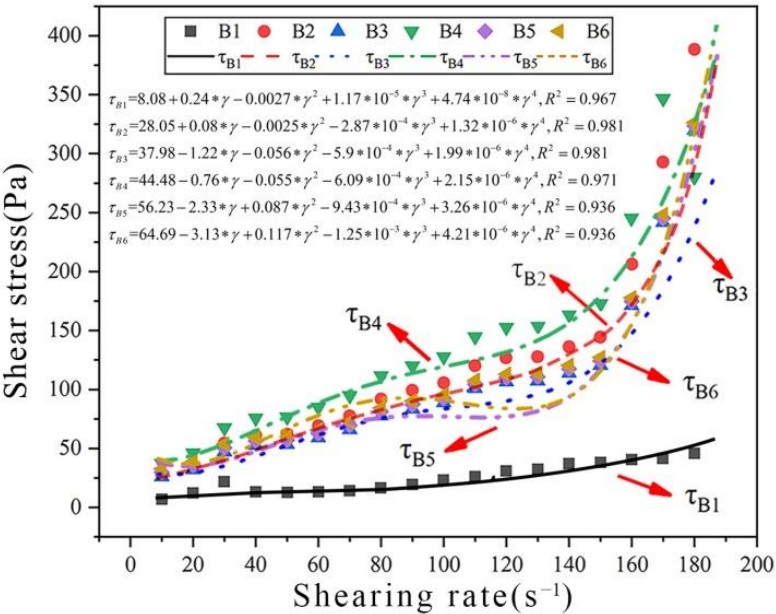

**Figure 7.** New rheological curve-fitting parameters of cement slurry carbonated under ultrasonic agitation.

**Table 8.** Rheological-model parameters of cement slurry carbonated under ultrasonic vibration.

| No. | $CO_2$ AA | $\tau_0$ (Pa) | $\varepsilon$ | $\eta$ | $\xi$ | $\leq$ | $R^2$ |
|-----|-----------|---------------|---------------|--------|-------|--------|-------|
| B1 | 0% | 8.08 | 0.24 | −0.0027 | $1.17 \times 10^{-5}$ | $4.74 \times 10^{-8}$ | 0.967 |
| B2 | 0.44% | 28.05 | 0.08 | −0.0025 | $-2.87 \times 10^{-4}$ | $1.32 \times 10^{-6}$ | 0.981 |
| B3 | 0.88% | 37.98 | −1.22 | −0.0560 | $-5.9 \times 10^{-4}$ | $1.99 \times 10^{-6}$ | 0.981 |
| B4 | 1.32% | 44.48 | −0.76 | −0.0550 | $-6.09 \times 10^{-4}$ | $2.15 \times 10^{-6}$ | 0.971 |
| B5 | 1.76% | 56.23 | −2.33 | 0.087 | $-9.43 \times 10^{-4}$ | $3.26 \times 10^{-6}$ | 0.936 |
| B6 | 2.20% | 64.69 | −3.13 | 0.117 | $-1.25 \times 10^{-3}$ | $4.21 \times 10^{-6}$ | 0.936 |

*5.4. Comparative Analysis of the Rheological Properties of Carbonated Cement Paste*

(1) Comparative analysis of viscosity

Viscosity curves of net cement pastes after carbonation under mechanical and ultrasonic vibration are shown in Figure 8a,b. Tables 9 and 10 show the viscosity values more clearly. The viscosity values were estimated for cement-paste shear rates of 30 L/s, 60 L/s, 90 L/s, 120 L/s and 150 L/s and $CO_2$ AA values of 0%, 0.44%, 0.88%, 1.32%, 1.76% and 2.20%, respectively.

It can be seen from Tables 9 and 10 that viscosity of cement paste before carbonation at shear rates of 30 L/s, 60 L/s, 90 L/s, 120 L/s and 150 L/s were 96.7%, 80%, 28.1%, 16.7% and 41.5% higher than those carbonated under mechanical vibration. At $CO_2$ AA = 0.44%, the viscosities under ultrasonic vibration were 19.6%, 52.6%, 28.6%, 32.6% and 67.3% higher than under mechanical vibration. At $CO_2$ AA = 0.88%, the viscosities under ultrasonic vibration were 103.8%, 96.2%, 58.8%, 19.2% and 15.4% higher than under mechanical vibration. At $CO_2$ AA = 1.32%, the viscosities under ultrasonic vibration were 83.3%, 83.3%, 5.3%, 19.4% and 0% higher than under mechanical vibration. At $CO_2$ AA = 1.76%, the viscosities under ultrasonic vibration were 48%, 26.3%, 16.9%, 98.2% and 87.5%. At $CO_2$

AA = 2.20%, the viscosities under ultrasonic vibration were 80.9%, 64.2%, 55.8%, 141.5% and 136.5% higher.

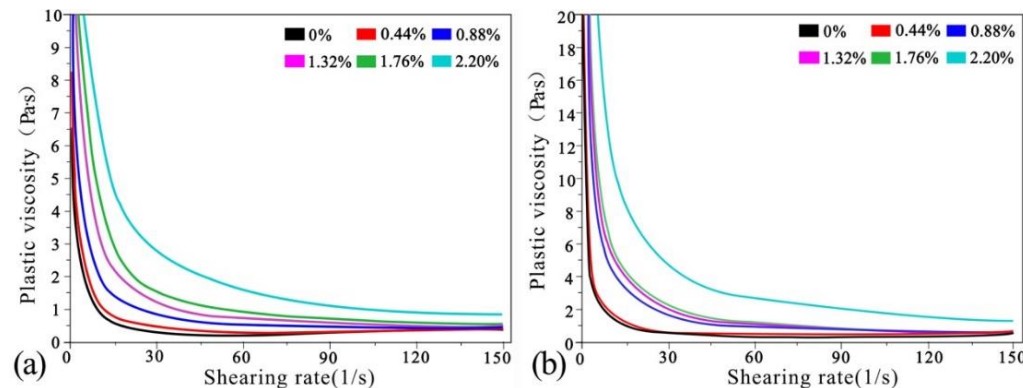

**Figure 8.** Viscosity curves of cement paste carbonated by (**a**) mechanical and (**b**) ultrasonic vibration.

**Table 9.** Viscosity values of net cement pastes carbonated under mechanical vibration (Pa·s).

| $CO_2$ AA/% | 30 L/s | 60 L/s | 90 L/s | 120 L/s | 150 L/s |
|---|---|---|---|---|---|
| 0 | 0.3 ± 0.11 | 0.2 ± 0.08 | 0.32 ± 0.13 | 0.36 ± 0.05 | 0.41 ± 0.06 |
| 0.44 | 0.51 ± 0.12 | 0.38 ± 0.11 | 0.42 ± 0.08 | 0.43 ± 0.06 | 0.55 ± 0.08 |
| 0.88 | 0.8 ± 0.13 | 0.52 ± 0.12 | 0.51 ± 0.07 | 0.52 ± 0.12 | 0.52 ± 0.09 |
| 1.32 | 1.2 ± 0.35 | 0.72 ± 0.23 | 0.64 ± 0.13 | 0.62 ± 0.13 | 0.62 ± 0.13 |
| 1.76 | 1.5 ± 0.45 | 0.95 ± 0.34 | 0.71 ± 0.07 | 0.65 ± 0.15 | 0.63 ± 0.15 |
| 2.20 | 2.72 ± 0.58 | 1.51 ± 0.56 | 1.15 ± 0.35 | 0.95 ± 0.34 | 0.83 ± 0.16 |

**Table 10.** Viscosity values of net cement pastes carbonated under ultrasonic vibration (Pa·s).

| $CO_2$ AA/% | 30 L/s | 60 L/s | 90 L/s | 120 L/s | 150 L/s |
|---|---|---|---|---|---|
| 0 | 0.59 ± 0.15 | 0.36 ± 0.11 | 0.41 ± 0.15 | 0.42 ± 0.18 | 0.58 ± 0.12 |
| 0.44 | 0.61 ± 0.16 | 0.58 ± 0.12 | 0.54 ± 0.12 | 0.57 ± 0.12 | 0.92 ± 0.34 |
| 0.88 | 1.63 ± 0.47 | 1.02 ± 0.35 | 0.81 ± 0.23 | 0.62 ± 0.12 | 0.6 ± 0.22 |
| 1.32 | 2.2 ± 0.51 | 1.32 ± 0.35 | 0.93 ± 0.34 | 0.74 ± 0.23 | 0.62 ± 0.22 |
| 1.76 | 2.22 ± 0.52 | 1.2 ± 0.55 | 0.83 ± 0.33 | 1.57 ± 0.66 | 1.49 ± 0.66 |
| 2.20 | 4.71 ± 0.82 | 2.62 ± 0.82 | 2.15 ± 0.81 | 1.55 ± 0.56 | 1.38 ± 0.56 |

Comparison of the two vibration modes shows that the viscosity after ultrasonic vibration is higher than that under mechanical vibration, indicating that ultrasonic vibration can effectively increase the viscosity of fresh cement paste. Moreover, the viscosity of cement paste increases gradually with increases in $CO_2$ AA under both vibration modes. At $CO_2$ AA = 0.44%, the viscosity changes the least with increases in shear rate. At $CO_2$ AA = 0.88%, 1.32% and 1.76%, the viscosity presents similar variations but increases to some extent. At $CO_2$ AA = 2.20%, the viscosity value changes the most.

(2) Comparative analysis of yielding stress

The yield stress of cement paste carbonated under ultrasonic vibration, which was derived from the ultrasonic rheological model, is shown in Figure 9. Mechanical vibration was used as the control group. The yield stress under mechanical vibration was derived from the H-B model. It can be seen from Figure 9 that $CO_2$ AA = 0%, 0.44%, 0.88%, 1.32%, 1.76% and 2.20%. At these values, the yield stresses of cement paste under ultrasonic vibration were 12.2%, 32.7%, 45.1%, 46.3%, 26.5% and 31.3%, respectively, which were higher than those under mechanical vibration. Under ultrasonic vibration, the yield stresses were 247%, 35.4.%, 17.1%, 26.4% and 31.3% higher, respectively. According to the measured data, ultrasonic vibration can effectively improve the yield stress of carbonated cement paste. The yield stress of carbonated cement paste is positively related to $CO_2$ AA.

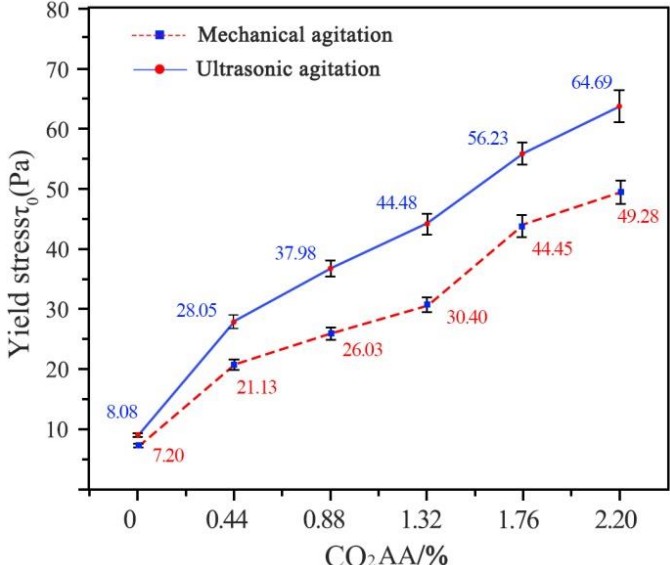

**Figure 9.** Yield stress of cement paste carbonated under ultrasonic vibration.

### 5.5. Effects of PS on the Rheological Properties of Cement Paste Carbonated under Ultrasonic Vibration

(1) Analysis of shear stress variation

Rheological curves of fresh carbonated cement pastes before and after PS addition are shown in Figure 10. The curves of shear stress after carbonation under ultrasonic vibration are shown in Figure 10a. The curves of shear stress after $CO_2$ carbonation and PS addition are shown in Figure 10b. Table 11 was plotted according to the rheological curves in Figure 10b. The variations in shear stress before adding PS can refer to the comparative analysis in Table 5.

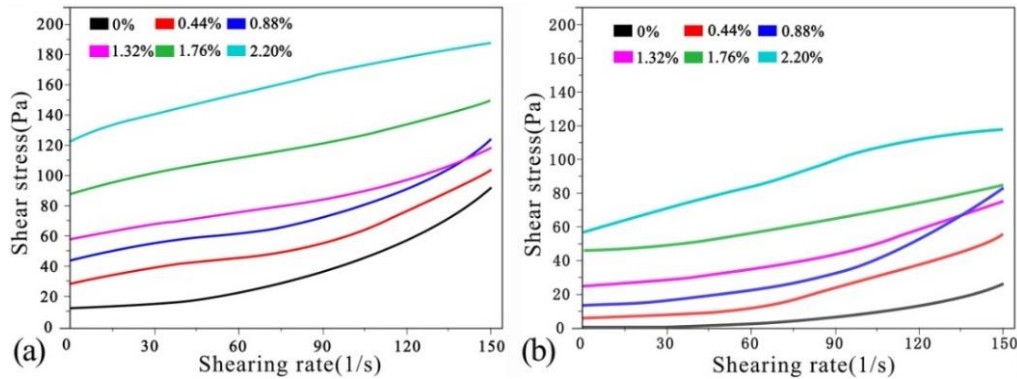

**Figure 10.** Rheological curves of carbonated fresh cement paste (**a**) before and (**b**) after adding water reducer.

**Table 11.** Shear stress of fresh carbonated cement paste after adding PS (Pa).

| $CO_2$ AA% | 30 L/s | 60 L/s | 90 L/s | 120 L/s | 150 L/s |
|---|---|---|---|---|---|
| 0 | $1 \pm 0.045$ | $3.1 \pm 0.14$ | $6.8 \pm 0.31$ | $13.5 \pm 0.59$ | $26.2 \pm 1.15$ |
| 0.44 | $8.1 \pm 0.45$ | $12.2 \pm 0.58$ | $24.4 \pm 1.19$ | $38.1 \pm 1.74$ | $56.3 \pm 2.26$ |
| 0.88 | $16.2 \pm 0.81$ | $20.5 \pm 0.95$ | $34.3 \pm 1.63$ | $51.4 \pm 2.33$ | $83.9 \pm 4.02$ |
| 1.32 | $27.2 \pm 1.24$ | $35.5 \pm 1.66$ | $44.3 \pm 2.18$ | $59.3 \pm 2.46$ | $75.5 \pm 3.49$ |
| 1.76 | $49.9 \pm 2.09$ | $58.1 \pm 2.72$ | $65.2 \pm 3.15$ | $75.2 \pm 3.15$ | $86.4 \pm 4.18$ |
| 2.20 | $72.2 \pm 3.15$ | $84.1 \pm 4.07$ | $100 \pm 4.45$ | $110.8 \pm 5.01$ | $118.1 \pm 5.11$ |

The shear stresses of fresh cement pastes before and after adding water reducer are compared next. At shear rates of 30 L/s, 60 L/s, 90 L/s, 120 L/s and 150 L/s and before carbonation, the shear stresses of fresh cement pastes with added water reducer were 92.2%, 86.2%, 81.9%, 76.4% and 71.5% lower than those without water reducer. At $CO_2$ AA = 0.44%, the shear stresses of cement pastes were 79.8%, 73.6%, 55.6%, 50.6% and 83.3% lower, respectively. At $CO_2$ AA = 0.88%, the shear stresses of cement pastes were 70.5%, 67.2%, 53.6%, 41.5% and 32.6% lower, respectively. At $CO_2$ AA = 1.32%, the shear stresses were 61.2%, 54.3%, 47.9%, 39.1% and 36.9% lower, respectively. At $CO_2$ AA = 1.76%, the shear stresses were 52.5%, 47.9%, 46.6%, 44.4% and 42.4% lower. At $CO_2$ AA = 2.20%, the shear stresses were 48.6%, 45.6%, 40.3%, 37.6% and 37% lower, respectively.

Based on the above comparison, the shear stress of carbonated fresh cement paste declines significantly after adding water reducer. Even in fresh cement pastes with the same amount of $CO_2$, the shear stresses of cement pastes with and without water reducer decline sharply.

(2) Analysis of viscosity-curve variation

Viscosity curves of fresh carbonated cement pastes before and after PS addition are shown in Figure 11. The curves for pastes carbonated under ultrasonic vibration are shown in Figure 11a. The viscosity curves for carbonated pastes after PS addition are shown in Figure 12b. Table 12 is plotted according to the viscosity curves after adding water reducer (Figure 11b). The variation in viscosity before PS addition can be compared in Table 10.

The viscosities of fresh cement pastes before and after PS addition are compared in this section. Without a supply of $CO_2$ and at shear rates of 30 L/s, 60 L/s, 90 L/s, 120 L/s and 150 L/s, the viscosities of cement pastes decreased by 74.6%, 55.6%, 58.5%, 61.9% and 67.2%, respectively, after PS addition. At $CO_2$ AA = 0.44%, the viscosities decreased by 77.9%, 80.4%, 67.9%, 51.6% and 36.7%. At $CO_2$ AA = 0.88%, the viscosities decreased by 77.9%, 80.4%, 67.9%, 51.6% and 36.7%. As $CO_2$ AA continued to increase to 1.32%, the viscosities decreased by 61.4%, 64.4%, 52.7%, 33.8% and 11.3%. When $CO_2$ AA = 1.76%, the viscosities decreased by 86%, 43.3%, 26.5%, 61.8% and 59.1%. At $CO_2$ AA = 2.20%, the viscosities decreased by 55.2%, 57.3%, 50.7%, 43.2% and 34.8%, respectively.

Through the above comparison, adding water reducer can decrease the viscosity of carbonated cement paste effectively. The viscosity of fresh cement paste increased to some extent with increases in $CO_2$ AA. Hence, it is concluded that adding water reducer can effectively improve the liquidity of cement paste after carbonation. However, such liquidity weakens with increases in $CO_2$ AA.

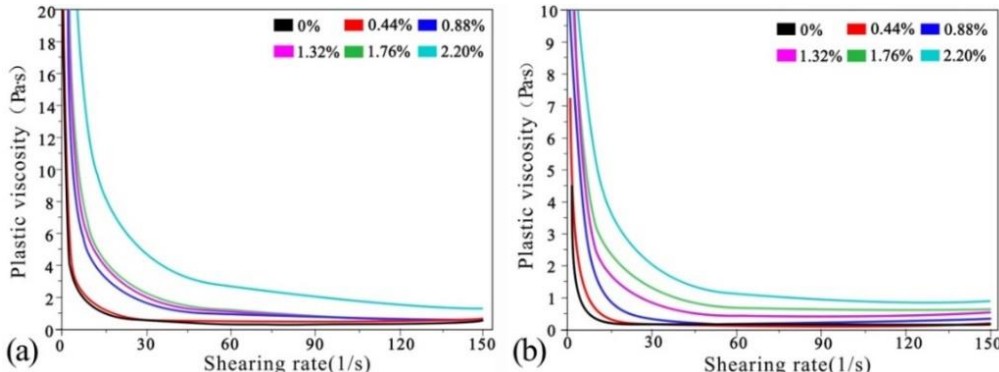

**Figure 11.** Viscosity curves of fresh carbonated cement pastes (**a**) before and (**b**) after adding water reducer.

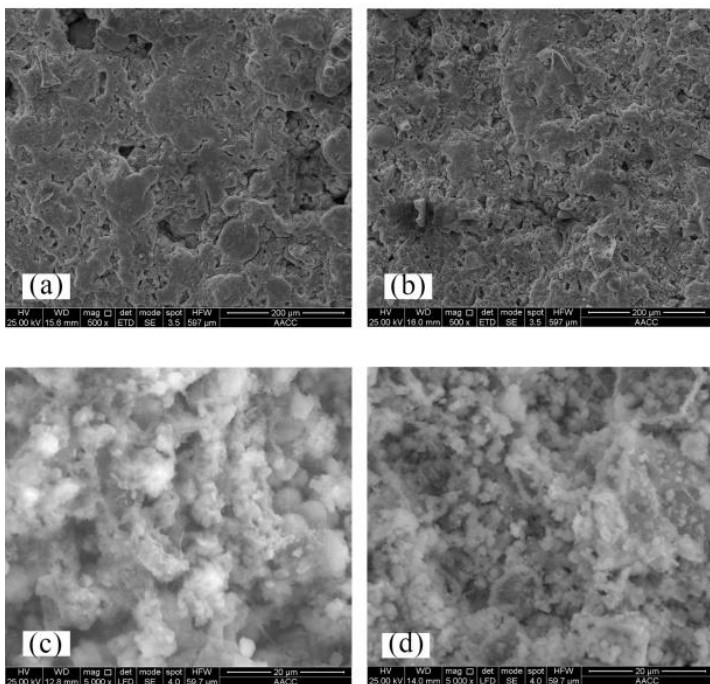

**Figure 12.** SEM images of cement pastes before and after adding water reducer: (**a**) before (500×); (**b**) after (500×) (**c**) before (5000×); (**d**) after (5000×).

**Table 12.** Viscosity values of fresh carbonated cement pastes before and after PS addition (Pa·s).

| CO$_2$ AA/% | 30 L/s | 60 L/s | 90 L/s | 120 L/s | 150 L/s |
|---|---|---|---|---|---|
| 0 | 0.15 ± 0.05 | 0.16 ± 0.04 | 0.17 ± 0.06 | 0.16 ± 0.08 | 0.19 ± 0.08 |
| 0.44 | 0.19 ± 0.07 | 0.12 ± 0.05 | 0.13 ± 0.05 | 0.14 ± 0.06 | 0.2 ± 0.09 |
| 0.88 | 0.36 ± 0.16 | 0.2 ± 0.09 | 0.26 ± 0.06 | 0.3 ± 0.17 | 0.38 ± 0.11 |
| 1.32 | 0.85 ± 0.03 | 0.47 ± 0.12 | 0.44 ± 0.08 | 0.49 ± 0.12 | 0.55 ± 0.12 |
| 1.76 | 1.31 ± 0.07 | 0.68 ± 0.13 | 0.63 ± 0.12 | 0.6 ± 0.229 | 0.61 ± 0.13 |
| 2.20 | 2.11 ± 0.09 | 1.12 ± 0.05 | 1.06 ± 0.35 | 0.88 ± 0.23 | 0.90 ± 0.26 |

## 6. Microstructural Analysis of Cement Paste with Water Reducer

### 6.1. Microstructural Characterisation of Cement Paste with Water Reducer

Scanning electron microscope (SEM) images of hydrated cement paste are shown in Figure 12. SEM images of hydrated cement pastes (CO$_2$ AA is 1.76%) before and after water reducer addition are shown in Figure 12a,b (the enlargement factor is 500×). It can be seen from Figure 12a that before adding water reducer, flocculants of cement paste show relatively complicated distributions and the diameter-size distribution differs greatly (diameter = 10–50 μm).

It can be seen from Figure 12b that for cement paste with water reducer, the flocculant distribution is relatively uniform and the particle size also decreases. The flocculant particles become tighter. Finally, it can be seen from Figure 12a,b that porosity decreases significantly after adding water reducer.

SEM images at 5000× magnification before and after adding water reducer are shown in Figure 12c,d. Smaller-sized hydration structures can be clearly observed. It can be seen from Figure 12c that cement-particle sizes are uneven before adding water reducer and flocculants look like the shape of the cotton, accompanied by high porosity. There are needle-like AFt crystals produced by the hydration of C$_3$A, as well as CaCO$_3$ crystals formed by carbonation of the cement paste.

Additionally, it can be seen from Figure 12d that the cement-particle distribution with water reducer is more uniform and the pores are smaller. The main sizes in the system change. Except for some large cement particles, particles 2–4 μm in diameter dominate.

The particle size decreases and the structure is relatively loose. The results demonstrate that after adding water reducer, flocculation particles are dispersed effectively and the liquidity of the cement paste increases accordingly.

### 6.2. Rheological Microstructural Model of Cement Paste with Water Reducer after Carbonation

(1) Construction of the rheological microstructure model

The rheological microstructural model of cement paste with water reducer after carbonation is shown in Figure 13. Adding water reducer to fresh cement paste not only prolongs the chemical reaction of the cement effectively but also changes the volume and space distributions of the products, thus decelerating coagulation after hydration of the cement. Moreover, water-reducer molecules adhering to particle surfaces may induce changes in electrical properties. Influenced by "steric hindrance", flocculates disperse and flocculation water is released, thus increasing free water [29,30]. According to the mechanism of the water reducer and combined with the above rheological analysis, a rheological microstructural model of paste with water reducer can be derived.

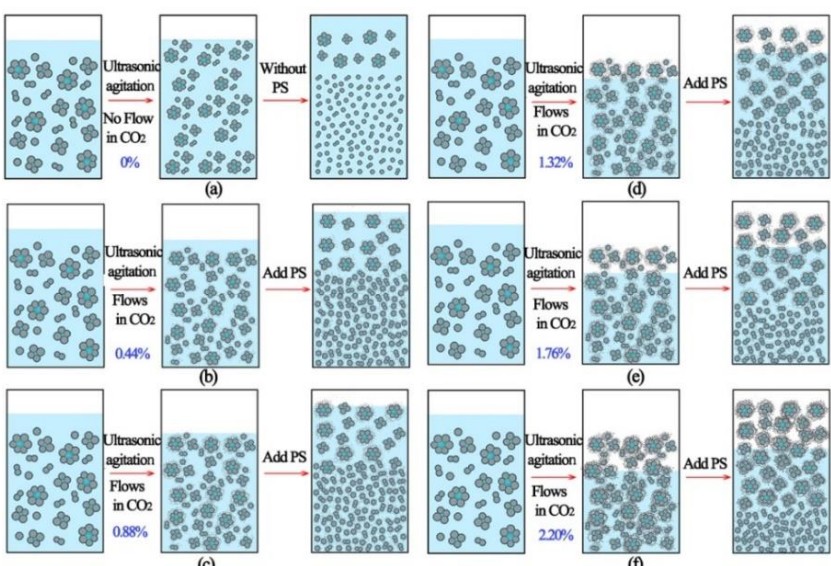

**Figure 13.** Rheological model of cement pastes with water reducer at (**a**) 0% $CO_2$; (**b**) 0.44% $CO_2$; (**c**) 0.88% $CO_2$; (**d**) 1.32% $CO_2$; (**e**) 1.76% $CO_2$; (**f**) 2.20% $CO_2$.

Since water reducer can delay the hydration of cement, it not only decelerates the coagulation of cement to some extent but also changes the volumetric and spatial distribution of hydration products and hinders the flocculation of cement particles. As a result, the liquidity of cement can be maintained to some extent in the early stage. According to the mechanism of the water reducer and with considerations of the above rheological analysis, the rheological microstructural model of paste with water reducer can be gained [31].

(2) Rheological model analysis of cement paste with water reducer after carbonation

Figure 13a shows that flocculants in cement paste without $CO_2$ disperse uniformly under ultrasonic vibration. After water reducer is added, flocculation water in flocculants is released and flocculants are dispersed into several small cement particles. Cement pastes supplied with 0.44%, 0.88%, 1.32%, 1.76% and 2.20% $CO_2$ are shown in Figure 14b–f. Clearly, $CO_2$ and cement hydrates react to produce $CaCO_3$ crystals, which adhere to surrounding flocculants. With increases in $CO_2$ AA in fresh cement paste, $CaCO_3$ crystals adhering to surrounding flocculants also increase continuously, thus restricting the effective release of free water from the fresh cement paste. Moreover, free-water release decreases with increases in $CO_2$ AA, thus making the liquidity of the cement paste decline gradually.

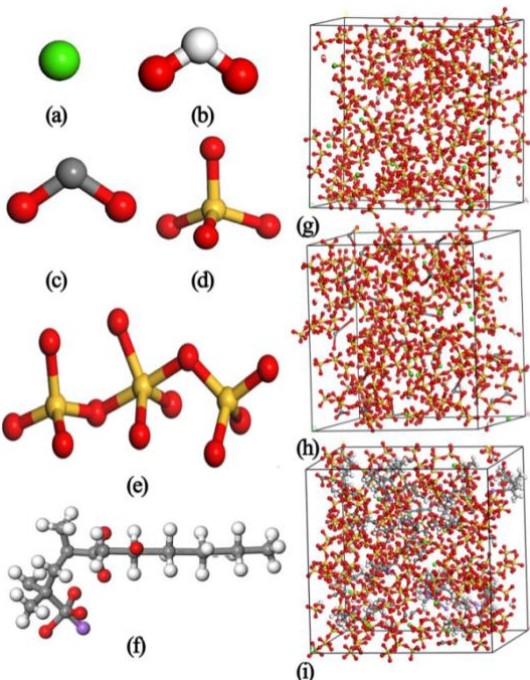

**Figure 14.** Model basic unit and build initial model: (**a**) Ca; (**b**) $H_2O$; (**c**) $CO_2$; (**d**) C-S-H-Q1; (**e**) C-S-H-Q2; (**f**) PS; (**g**) Molecular structure of PC; (**h**) Molecular structure of PC + $CO_2$; (**i**) Molecular structure of PC + $CO_2$.

Moreover, flocculants in fresh cement pastes are dispersed effectively under ultrasonic vibration after water reducer is added, thus enabling the release of water wrapped in the flocculants. Hence, the free water in fresh cement paste increases and the liquidity is improved. It can be seen from Figure 13b–f that with increases in $CO_2$ AA, the number of flocculating structures that are separated effectively decreases, thus resulting in a relatively small amount of released free water. The liquidity of cement paste declines continuously.

(3) Summary of the rheological model of carbonated cement paste with water reducer

The liquidity of cement paste with water reducer deteriorates after $CO_2$ absorption. This is mainly because with increases in $CO_2$ AA, the $CaCO_3$ crystals on the surfaces of flocculants and cement particles increase, including AFt crystals produced by the hydration of cement [32]. After water reducer is added, water-reducer molecules enter organic mineral phases. Since excessive $CaCO_3$ crystals and AFt are wrapped on the surfaces of flocculants and cement particles, water-reducer molecules cannot develop the role well. During vibration, only some flocculants are dispersed, thus preventing flocculant water from being released well. Accordingly, free-water growth declines. As a result, the liquidity of cement paste worsens after $CO_2$ absorption. In a word, the liquidity of cement paste is negatively related to its $CO_2$ AA.

## 7. Molecular Analysis of the C-S-H Gel Structure of Absorption $CO_2$ Cement Slurry

A COMPASS force field was applied to the C-S-H gel structure using the Amorphous Cell module in Materials Studio. The selected molecules were input to the model through the constructed PC model to analyse the kinetic energies and pressures in the molecules as well as the radial distribution function [33].

### 7.1. Construction of the Molecular Model

Ca, $H_2O$, $CO_2$, C-S-H-Q1, C-S-H-Q2 and PS were chosen as the basic units (Figure 14a–f). According to the molecular proportions of the proposed C-S-H gel, 100 Poly (sialate-siloxo) molecules were chosen randomly. A total of 50 Na+ and 20 $H_2O$ molecules were distributed into the simulation box, thus establishing an initial structural model of C-S-H gel (PC, Figure 14g). Next, 40 $CO_2$ molecules were put into the constructed PC model to establish a

model in which $CO_2$ was supplied to the C-S-H gel (PC + $CO_2$, Figure 14h). In the PC + $CO_2$ model, 20 water-reducer molecules were added to establish a PC + $CO_2$ + water-reducer model (PC + $CO_2$ + JS, Figure 14i).

### 7.2. Kinetic-Energy and Potential-Energy Variations in Molecules

The influences of molecular kinetic energy and potential energy are shown in Figure 15. Clearly, the potential and kinetic energies of the molecular structures of PC, pure cement with $CO_2$ (PC + $CO_2$) and pure cement with $CO_2$ and water reducer (PC + $CO_2$ + JS) are positive. This indicates that interactive intermolecular forces in the NASH gel structure create positive work and the forces among molecules are repulsive. The molecular potential energies of the C-S-H gel, C-S-H gel + $CO_2$ and C-S-H gel + $CO_2$ + water reducer are 13,113.809 kJ/mole, 13,570.697 kJ/mole and 22,864.47 kJ/mole, respectively. This indicates that the intermolecular repulsion forces in the C-S-H gel + $CO_2$ and C-S-H gel + $CO_2$ + water reducer are larger than the attractive forces. The interaction forces among molecules make negative work and the molecular potential energy increases. The $CO_2$ and water reducer can increase repulsion among particles in the molecular structure, thus improving the dispersion effect of C-S-H gels.

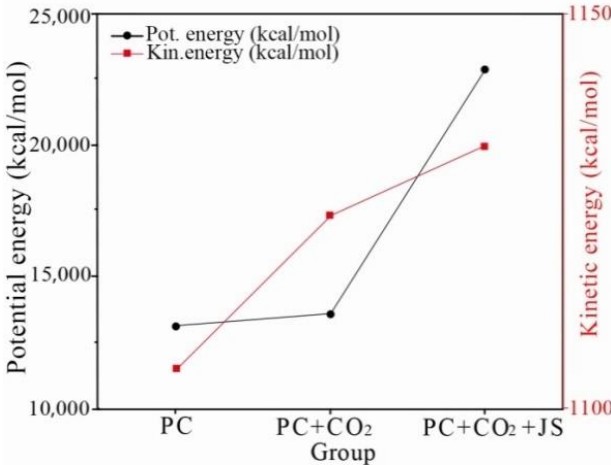

**Figure 15.** Changes in molecular kinetic energy and potential energy.

The molecular kinetic energies of the C-S-H gel, C-S-H gel + $CO_2$ and C-S-H gel + $CO_2$ + water reducer are 1104.896 kJ/mole, 1124.326 kJ/mole and 1133.174 kJ/mole, respectively. This indicates that adding $CO_2$ and $CO_2$ + CSH into a CSH gel can intensify the vibration amplitude of the molecular particles in the structure and thereby decrease the intermolecular forces. $CO_2$ and water reducer can increase the driving forces among the particles in the molecular structures, thus improving the dispersion effect of the CSH gel.

### 7.3. Internal Pressure in Molecules

The influences of $CO_2$ and water reducer on intermolecular pressures in the C-S-H gel structure are shown in Figure 16. Adding $CO_2$ and water reducer to the C-S-H gel can transform the intermolecular pressure from positive to negative, indicating that the C-S-H gel is generally transformed from compression to tension. The internal pressures in the PC, PC + $CO_2$ and PC + $CO_2$ + JS are 0.173 GPa, −0.054 GPa and −0.031 GPa, respectively. To sum up, adding $CO_2$ or $CO_2$ + water reducer can decrease the intermolecular constraining force and increase movement among molecules.

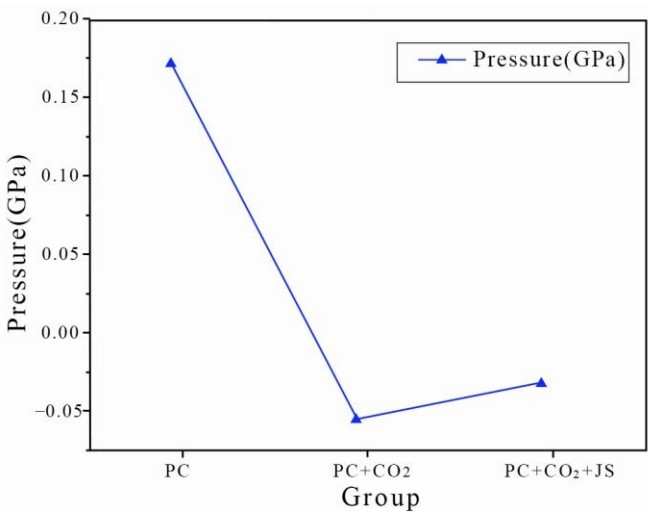

**Figure 16.** Variation in intermolecular pressure.

*7.4. Radial Distribution Function*

The radial distribution functions (RDFs) for PC, PC + $CO_2$ and PC + $CO_2$ + JS are shown in Figure 17. When the relative distance among particles is r > 3.0 Å, the numerical value of RDF approaches 1. When r < 1.07 Å, RDF = 0. This indicates that the nearest distance among the molecules in the calculated NASH system will be at least 1.07 Å, and the longest distance is at least 3.0 Å. The radii (arrived at 1.71 Å) of the maximum peaks in PC, PC + $CO_2$ and PC + $CO_2$ + JS are 33.3512 Å2, 31.1548 Å2 and 22.7978 Å2, respectively.

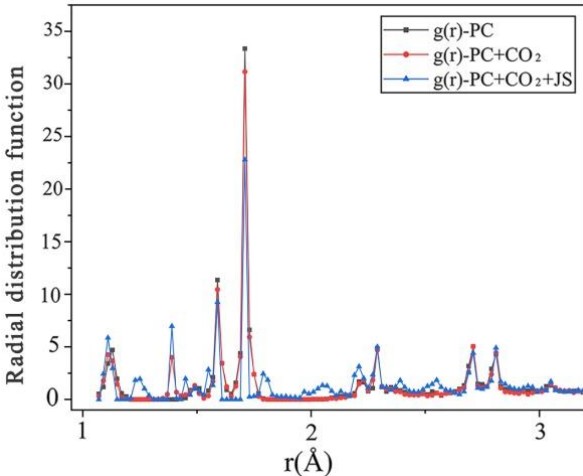

**Figure 17.** Radial distribution function.

In the range of RDF peaks, the probability of having other molecules is the highest and the local density is the highest. With increases in regional density, the atomic shells will deform due to the existence of a force field. Once the lattice structure of the new solid phase begins to expand and grow, the additional peak in the RDF will reflect the atomic effect [34]. This proves that the C-S-H gel structure is decomposed into movable nanoclusters to obtain high cohesive energies and maintain structural stability. Once $CO_2$ and water reducer are added, the gel groups cause shifts of the peaks through the dispersion effect.

Results analysis. According to the analysis of the variation in intramolecular pressure, the constraining forces among molecules are decreased by adding water reducer, which increases the movement among molecules. This can further decrease the constraining forces among molecules. The intramolecular repulsion in the C-S-H gel + $CO_2$ is greater than the attractive force. The interaction forces among molecules make negative charges

and the molecular potential energy increases. The $CO_2$ and water reducer can increase repulsion among the different particles in the molecular structures. In other words, the $CO_2$ and water reducer can increase the driving forces among the particles in the molecular structures, which can better improve the dispersion effect of the C-S-H gel. Furthermore, it is concluded from the RDF analysis that the C-S-H gel structure is decomposed into movable nanoclusters to obtain high cohesive energy and maintain high structural stability. Once the $CO_2$ and water reducer are added, the gel groups can cause shifts of the peaks due to the dispersion effect. This shows that the gel groups are effectively distributed, thus improving various aspects of the performance of the cement pastes.

## 8. Conclusions

Our previous research on the $CO_2$ absorption performance of fresh cement slurry demonstrated that ultrasonic vibration could more effectively improve $CO_2$ absorption than mechanical mixing. Moreover, the fluidity of the slurry decreased with the increase in $CO_2$ absorption. In addition, ultrasonic vibration can effectively improve the fluidity of cement slurry followed by $CO_2$ absorption [21]. If a water-reducing agent is added to the slurry, it not only increases its fluidity but also restores the working performance that has lost fluidity [12]. Therefore, this paper focused on the dynamic analysis through the principle of rheology and establishing the molecular model while observing the change in internal microstructure.

Under ultrasonic vibration, the fluidity of fresh paste decreases with the increase in $CO_2$ absorption and with the addition of a superplasticiser, the fluidity of the decreased slurry could be restored, which indicates an unclear principle of change. Therefore, in this paper, the rheological principle was analysed. Moreover, the internal microstructural change was observed, followed by the establishment of the rheological structure model. Through the development of a molecular model and dynamic analysis, the internal workability-change mechanism of fresh slurry along with superplasticiser was intensively explored while subjected to ultrasonic vibration. The results indicated that ultrasonic vibration can increase the shear stress and plastic viscosity of cement paste more effectively than the cement paste subjected to only $CO_2$ absorption via mechanical stirring. In addition, the shear stress and viscosity of cement paste could increase with the increase in $CO_2$ absorption. At the same time, the addition of polycarboxylic-acid superplasticiser to the cement paste along with $CO_2$ absorption can more effectively reduce the shear stress and plastic viscosity of the fresh paste while restoring and improving its fluidity. In addition, the rheological model of cement paste subjected to $CO_2$ absorption by mechanical stirring fitted well with the H-B rheological model, while the rheological model under ultrasonic vibration was within the range of the traditional rheological model. However, it does not completely accord with the actual rheological characteristics. Therefore, an ultrasonic rheological model was proposed for the cement-based materials which could suitably absorb $CO_2$ under ultrasonic vibration. In addition, both $CO_2$ and superplasticiser could reduce the intermolecular binding force and increase the driving force between the intermolecular particles. As a result, the dispersion effect of C-S-H gel could enhance, if the gel groups are effectively distributed. Finally, the rheological properties of the slurry could improve.

Due to the lack of research on the absorption of $CO_2$ by cement slurry and the limitations of experimental equipment, at present, we are only able to report the $CO_2$ absorption improvement process and its impact on fluidity. The next step will be investigating the effect of $CO_2$ absorption on the carbonation resistance of concrete and its corresponding mechanism.

**Author Contributions:** Study design, Y.J.; conduct of the study, L.L. (Lili Liu); conceptualisation, J.Z.; data collection, L.L. (Longhai Li); methodology, L.L. (Lili Liu); writing—original draft preparation, L.L. (Lili Liu) and Y.J. All authors have read and agreed to the published version of the manuscript.

**Funding:** This word was funded by the National Natural Science Foundation of China (51972337) and Science and Technology Innovation Project of Xuzhou (Grant No. KC21014).

**Institutional Review Board Statement:** Not applicable.

**Informed Consent Statement:** Not applicable.

**Data Availability Statement:** Not applicable.

**Acknowledgments:** The authors would like to thank the technical staff of the Jiangsu Key Laboratory Environmental Impact and Structural Safety in Engineering for their technical support.

**Conflicts of Interest:** The authors declare no conflict of interest.

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
