# Peer review of "Influences of CO2 Absorption under Ultrasonic Vibration and Water-Reducer Addition on the Rheological Properties of Cement Paste"

_applsci, doi:10.3390/app12083739_

Round 1
Reviewer 1 Report
In this study, the authors studied the influences of ultrasonic vibration and water reducer addition on the rheological properties of cement pastes after absorbing various amounts of CO2. Also they investigated the rheological properties of yield stress and plastic viscosity (PV).
The paper is well-organized. All sections are completed and enough information has been provided.
Authors need to include more data about the current and previous work in this field in the introduction section. They need to clearly highlight the novelty in this study and com[are it clearly with previous works done by other research groups.
The statistical analysis need to be added including number of samples and all error bar and accuracy in the graphs/table.
Some information in some images are not legible and need to be revised/replaced.
It can be published after making the indicated minor changes.
Author Response
Dear Reviewer,
We really appreciate the reviewer for the time and effort in reviewing our manuscript. The comments are very impressive and valuable to improve the manuscript. With our best efforts and knowledge, we have accordingly edited and revised the manuscript based on comments. The following are our response to the comments from the reviewers, and the revisions were marked in red in the paper.
Thanks for your time. Looking forward to hearing from you.
Yours sincerely
Review #1:
Point 1: In this study, the authors studied the influences of ultrasonic vibration and water reducer addition on the rheological properties of cement pastes after absorbing various amounts of CO2. Also they investigated the rheological properties of yield stress and plastic viscosity (PV).
The paper is well-organized. All sections are completed and enough information has been provided.
Authors need to include more data about the current and previous work in this field in the introduction section. They need to clearly highlight the novelty in this study and com[are it clearly with previous works done by other research groups.
Response 1: Thanks for your kind guidance. We have rewritten the introduction and conclusion sections in the revised version.
Point 2: The statistical analysis need to be added including number of samples and all error bar and accuracy in the graphs/table.
Response 2: Thank you for pointing out the deficiencies. The statistical analysis were added in the related graphs/table (Table 4, Table 5, Table 9, Table 10, Table 11, Table 12 and Figure 9).
Point 2: Some information in some images are not legible and need to be revised/replaced.It can be published after making the indicated minor changes.
Response 3: Thank you for the thoughtful advice. The related images were revised in the revised version (Figure 2、3、4、5、8、9、10、11、13、15、16、17).

Reviewer 2 Report
- Abstract needs to be improved.
- Literature review is very shallow and is not up-to-date, please enrich this section further.
- The gap, significance and objective of the study are missing.
- Figures have a low resolution.
- Critical analysis and discussion are highly required to be compared with previous studies.
- The conclusion is very poor and it has no standing form.
- The current writing form of the conclusion is superficial and shallow. It must be improved in highlight points supported with values/percentages.
- Recommend the potential applications of the product of this study.
- Suggest further studies and highlight the limitations of this study.
- The format of the references is not accurate.
Author Response
Dear Reviewer,
We really appreciate the reviewer for the time and effort in reviewing our manuscript. The comments are very impressive and valuable to improve the manuscript. With our best efforts and knowledge, we have accordingly edited and revised the manuscript based on comments. The following are our response to the comments from the reviewers, and the revisions were marked in red in the paper.
Thanks for your time. Looking forward to hearing from you.
Yours sincerely
Review #2:
Point 1:Abstract needs to be improved.
Response 1: Thanks for your careful review. We have rewritten the abstract in the revised version.
Point 2:Literature review is very shallow and is not up-to-date, please enrich this section further.
Response 2: Thanks for your kind guidance. We have rewritten the introduction section in the revised version.
Point 3:The gap, significance and objective of the study are missing.
Response 3: Thank you for pointing out our mistake. We have added some descriptions in the revised version.
Point 4:Figures have a low resolution.
Response 4: Thank you for the thoughtful advice. The related images were revised in the revised version (Figure 2、3、4、5、8、9、10、11、13、15、16、17).
Point 5:Critical analysis and discussion are highly required to be compared with previous studies.
Response 5: Thank you for the thoughtful suggestion and it can really make the paper more logical and complete. We have added some descriptions in introduction and conclusion section.
Point 6:The conclusion is very poor and it has no standing form.The current writing form of the conclusion is superficial and shallow. It must be improved in highlight points supported with values/percentages.Recommend the potential applications of the product of this study.Suggest further studies and highlight the limitations of this study.
Response 6: Sorry, we did not express this part properly. We have rewritten this section in the revised version.
Point 7:The format of the references is not accurate.
Response 7: Thanks for your careful review. We reviewed the references and corrected it in revised manuscript.

Reviewer 3 Report
The article describes an experimental study on the effect of ultrasound agitation on CO2 absorption by cement slurries.
The research presented in the article was correctly planned and conducted. They address an important and practical research issue.
The language and style of the article, especially in its second part, should be improved due to a large number of linguistic errors and shortcomings, for example in lines:
- 84,
- -units in line 144 and further,
- 156,
- 607, 608, 610 614 etc.
My main point that needs clarification concerns the similarity of the research carried out to that already published in Applied Sciences in 2021.
https://www.mdpi.com/2076-3417/11/15/6877
In my opinion, the given published studies need to be cited and demonstrate the novelty and additions of the presented topic in the light of the current publication.
Yours Sincerely,
Reviewer.
Author Response
Dear Reviewer,
We really appreciate the reviewer for the time and effort in reviewing our manuscript. The comments are very impressive and valuable to improve the manuscript. With our best efforts and knowledge, we have accordingly edited and revised the manuscript based on comments. The following are our response to the comments from the reviewers, and the revisions were marked in red in the paper.
Thanks for your time. Looking forward to hearing from you.
Yours sincerely
Review #3:
Point 1:The article describes an experimental study on the effect of ultrasound agitation on CO2 absorption by cement slurries.The research presented in the article was correctly planned and conducted. They address an important and practical research issue.
The language and style of the article, especially in its second part, should be improved due to a large number of linguistic errors and shortcomings, for example in lines:
84,
-units in line 144 and further,
156,
607, 608, 610 614 etc.
Response 1: Thank you for pointing out the deficiencies. We have carefully revised the manuscript according to the reviewers' comments, and also have re-scrutinized to improve the English by a language polishing service.
Point 2:My main point that needs clarification concerns the similarity of the research carried out to that already published in Applied Sciences in 2021.
https://www.mdpi.com/2076-3417/11/15/6877
In my opinion, the given published studies need to be cited and demonstrate the novelty and additions of the presented topic in the light of the current publication.
Response 2: Thanks for your gentle suggestion and it do make the narration is more comprehensive and reasonable. We have rewritten the introduction and conclusion section in the revised version.

Round 2
Reviewer 2 Report
The manuscript has been improved by almost the comments were addressed. except that there is a wrong format of the references. Thus, they should revise them accordingly.
Author Response
Point 1: The manuscript has been improved by almost the comments were addressed. except that there is a wrong format of the references. Thus, they should revise them accordingly..
Response 1: Thanks for your careful review. The format of the references were modified in the revised version.

Reviewer 3 Report
Most of my previous comments have been revised.
In the current version of the article, citations in places of corrections and additions should be adjusted, especially in the conclusion section.
Conclusions should also be reread, supplemented and made more specific.
Yours Sincerely,
Reviewer.
Author Response
Point 1: Most of my previous comments have been revised.In the current version of the article, citations in places of corrections and additions should be adjusted, especially in the conclusion section.Conclusions should also be reread, supplemented and made more specific.
Response 1: Thanks for your kind guidance. We have rewritten this section and added the related citations in the revised version.
